# RNA-Seq Reveals That Multiple Pathways Are Involved in Tuber Expansion in Tiger Nuts (*Cyperus esculentus* L.)

**DOI:** 10.3390/ijms25105100

**Published:** 2024-05-07

**Authors:** Guangshan Hou, Guojiang Wu, Huawu Jiang, Xue Bai, Yaping Chen

**Affiliations:** 1Key Laboratory of South China Agricultural Plant Molecular Analysis and Genetic Improvement and Guangdong Provincial Key Laboratory of Applied Botany, South China Botanical Garden, Chinese Academy of Sciences, Guangzhou 510650, China; hguangshan@163.com (G.H.); wugj@scib.ac.cn (G.W.); hwjiang@scib.ac.cn (H.J.); 2State Key Laboratory of Plant Diversity and Specialty Crops, South China Botanical Garden, Chinese Academy of Sciences, Guangzhou 510650, China; 3University of Chinese Academy of Sciences, Beijing 100049, China; 4Key Laboratory of Tropical Plant Resources and Sustainable Use, Xishuangbanna Tropical Botanical Garden, The Innovative Academy of Seed Design, Chinese Academy of Sciences, Menglun 666303, China; baixue2015@xtbg.ac.cn

**Keywords:** *Cyperus esculentus*, tuber expansion, transcriptomic analysis, phytohormone

## Abstract

The tiger nut (*Cyperus esculentus* L.) is a usable tuber and edible oil plant. The size of the tubers is a key trait that determines the yield and the mechanical harvesting of tiger nut tubers. However, little is known about the anatomical and molecular mechanisms of tuber expansion in tiger nut plants. This study conducted anatomical and comprehensive transcriptomics analyses of tiger nut tubers at the following days after sowing: 40 d (S1); 50 d (S2); 60 d (S3); 70 d (S4); 90 d (S5); and 110 d (S6). The results showed that, at the initiation stage of a tiger nut tuber (S1), the primary thickening meristem (PTM) surrounded the periphery of the stele and was initially responsible for the proliferation of parenchyma cells of the cortex (before S1) and then the stele (S2–S3). The increase in cell size of the parenchyma cells occurred mainly from S1 to S3 in the cortex and from S3 to S4 in the stele. A total of 12,472 differentially expressed genes (DEGs) were expressed to a greater extent in the S1–S3 phase than in S4–S6 phase. DEGs related to tuber expansion were involved in cell wall modification, vesicle transport, cell membrane components, cell division, the regulation of plant hormone levels, signal transduction, and metabolism. DEGs involved in the biosynthesis and the signaling of indole-3-acetic acid (IAA) and jasmonic acid (JA) were expressed highly in S1–S3. The endogenous changes in IAA and JAs during tuber development showed that the highest concentrations were found at S1 and S1–S3, respectively. In addition, several DEGs were related to brassinosteroid (BR) signaling and the G-protein, MAPK, and ubiquitin–proteasome pathways, suggesting that these signaling pathways have roles in the tuber expansion of tiger nut. Finally, we come to the conclusion that the cortex development preceding stele development in tiger nut tubers. The auxin signaling pathway promotes the division of cortical cells, while the jasmonic acid pathway, brassinosteroid signaling, G-protein pathway, MAPK pathway, and ubiquitin protein pathway regulate cell division and the expansion of the tuber cortex and stele. This finding will facilitate searches for genes that influence tuber expansion and the regulatory networks in developing tubers.

## 1. Introduction

The size of storage organs such as seeds, stems, tubers, and tuberous roots and the leaf size of vegetable species are the key traits that determine the economic yield of crops [1]. Because of the importance of plants for food and renewable industry sources, the mechanisms involved in organ size control are a longstanding biological question. At a cellular level, the cell number and size are two essential factors that determine the size of an organ. The cell number is dependent on the initial primordium, with the rate and the duration of cell division based on the activity of a series of proteins involved in DNA replication and the mitosis and control of the cell cycle. There is evidence that cell enlargement is based on turgor-driven cell expansion, cell wall loosening, and the de novo synthesis of cell wall components [2]. The cell division and enlargement processes of organs are, therefore, regulated by complex signaling pathways.

A number of genes related to organ size have been identified in different plants and are spread across these regulatory pathways. Seed size is determined by several signaling pathways that include the DA1 ubiquitin-proteasome, G-protein regulators, mitogen-activated protein kinase (MAPK), and miRNA regulators, HAIKU (IKU) pathways, and also phytohormone perception and homeostasis, and transcriptional regulators [3]. The DA1 ubiquitin-proteasome pathway is conserved in plants, with rice (*Oryza sativa*) having four DA1 homologous genes, including OsDA1, which has been shown to interact with OsUBP15, a positive regulator of grain size [4]. The heterotrimeric G-protein complex consists of Gα, Gβ, and Gγ sub-units that participate in the multiple processes of plant growth and development. A loss of function and the suppression of either Gα or Gβ has been shown to result in short grains in rice, suggesting that Gα and Gβ positively regulate grain length [5]. The MAPK cascades are composed of three types of protein kinases: the MAPK kinase kinase (MKKK), the MAPK kinase (MKK), and the MAPK [6]. These MAPK cascades are involved in regulating grain size in rice. The OsMKKK10-OsMKK4-OsMAPK6 cascade controls grain size in rice by promoting cell proliferation in the spikelet hull, while the IKU pathway controls grain size by influencing endosperm development, which has only been reported in canola (*Brassica napus*). In rice, the OsmiR396-OsGRF4-OsGIFs module has an important role in determining grain size control by mainly promoting cell enlargement and slightly promoting cell proliferation [7]. Brassinosteroid (BR) and indole-3-acetic acid (IAA) are the most important phytohormones involved in the maternal control of seed and grain size in both monocots and dicots. Phytohormones, related to signal transduction such as GS5, GSK2, and DLT/SMOS2, have been shown to regulate both cell division and cell enlargement [8].

Several molecular events have been shown to govern leaf organ size in Arabidopsis which include the above signaling pathways for the control of seed size. The DA1-ubiquitin-proteasome degradation, PEAPOD (PPD), KLUH (KLU), and GRF-GIF pathways, and some transcriptional regulators also regulate both cell division and expansion. PPD pathway genes are absent in all the Poaceae (monocot grasses) studied [9], while the KLU and GRF-GIF pathways have evolutionarily remained in Poaceae, such as maize (*Zea mays*), rice, and wheat (*Triticum aestivum*) [10,11]. The growth and the expansion of tuber and tuberous roots are regulated genetically, although the signaling pathways involved in this process remain unclear. Endogenous hormones play an important role in the process of tuber and tuberous root expansion. For example, the content of IAA increases gradually at the initial stage of root expansion in sweet potato (*Ipomoea batatas*) tuberous root, while in the potato (*Solanum tuberosum*), jasmonic acid (JA) in the tuber significantly promotes tuber initiation and bulking [12], with StJAZ1-like-mediated JA signaling also having an essential role in potato tuberization [13].

Tiger nuts are a grass-like perennial herb that belong to the sedge family Cyperaceae that is widely distributed in tropical, subtropical, and temperate zones. It can be used as an annual cultivated crop in agricultural production [14]. The tubers of the tiger nut not only contain a large amount of starch, oil, and carbohydrate, but also contain dietary fiber, vitamin C, vitamin E, flavonoids, and minerals [15]. Tiger nut oil is rich in unsaturated fatty acids, including oleic acid (64–68%), which have beneficial effects in patients with diabetes, cardiovascular disease, or obesity [16]. In recent years, several studies have used global transcriptome analysis to focus on the metabolism of sugars and the accumulation of reserve oil and starch in developing tiger nut tubers [17]. In addition, draft genomic sequences of tiger nut were reported recently [18]. This research provided the basic data for further molecular mechanism and genic studies on various growth and developing processes in the tiger nut tubers. The size of tubes is a key trait that determines the yield and mechanical harvesting of tiger nut tubes. However, only a small number of studies related to tubers development have been reported, and no specific pathways related to tuber expansion of tiger nut plants have been identified. Clarifying the mechanisms underlying tubers development is important for improving the tiger nut yield. Therefore, based on the observation of tuber anatomy and transcriptome sequencing during tuber development, the regulatory mechanism of tuber enlargement was discussed in this paper. First, we observed changes in the tubers during the developmental process of “B63”, an annual local variety grown in Yinchuan, in the Ningxia Hui Autonomous Region, China. The tuber expansion process of the tiger nut plant was divided into six stages, based on morphological indices and dry weight changes. Next, the fine processes of cell proliferation and cell enlargement of tiger nut tuber expansion was chartered according to their anatomical structure. Finally, we compared the differences in transcriptomes of the six different developmental stages, and investigated the gene regulatory networks associated with tiger nut tuber expansion to identify the key genes responsible for this growth. Our findings provide new insights into the complex mechanisms of tuber expansion and identified candidate target genes in tiger nut plants, which could be applied to the breeding of the high-yield tiger nut.

## 2. Results

### 2.1. Characteristics of Tuber Development

To establish a framework for developing the profiles of the tubers, we initially tested the changes in size and dry weight of tiger nut tubers at the following days after sowing: 40 d (S1); 50 d (S2); 60 d (S3); 70 d (S4); 90 d (S5); and 110 d (S6). Tubers developed at the tips of rhizomes at 40 d and reached the maximum size at 70 d. Accordingly, we defined three developmental phases of tubers in the tiger nut plant. The first phase was the tuber rapid expansion phase that occurred during the first 20 d (S1 to S3) after tuber initiation. The expansion rate in diameter was approximately 0.4 mm/d during this phase. The second phase included both the expansion and filling phases (S3 to S4). The tuber size reached a maximum at S4, with an expansion rate in the diameter of approximately 0.3 mm per day and an increase in the dry weight of approximately 62 mg per day during this phase. The third phase was the filling and mature phase (S4 to S6) (Figure 1).

The cross-sections in the S1 tubers showed that the stele contained five to six bundles of vascular bundles and the inward distribution of the xylem. Small cavities were formed by cell lysis that may have served as ventilation. A primary thickening meristem (PTM) surrounded the stele. In S2 tubers, the stele was nearly circular with five vascular bundles. The small cavity near the vascular bundles in a S1 tubers was filled with parenchyma cells and formed an irregularly shaped pith. An adventitious root formed at this stage. In S3 tubers, there was a clear boundary between the cortex and the stele formed by the procambium (PC) and PTM. The stele was filled mainly with pith parenchyma cells, with 15–20 vascular bundles distributed in a scattered manner. The boundaries between the pith and other parts of the stele became blurred. The anatomical structure of the tubers at S4 was similar to those at S3 (Figure 2). These results indicated that the cortical region was established at the S1 stage, while the transfer phase of the stele to create the pith parenchymal cell for storage occurred between S1 to S3.

Within the epidermis, the mature tiger nut tubers were composed of cortex and pith regions. In S1 tubers, the radius of the stele were relatively small, and the thickness of the cortex was 3.27 times greater than the radius of the stele. In S2 tubers, the thickness of the cortex of the tubers increased rapidly from S1 to S2, with the thickness of the cortex in S2 tubers being 5.04 times larger than the radius of the pith. From S2 to S3, the size of the pith increased rapidly, with the thickness of the cortex being 0.9 times of the radius of pith in tubers at S4. From S3 to S4, the tubers showed a synchronized increase in cortical thickness and pith radius (Table 1).

Increases in the size of the cortical and pith (or stele) regions in the tuber were primarily the result of cell division of the ground meristem of the PTM and cell enlargement. According to the cell size and cell layer number, cell division in the cortical region stopped between S1 and S2, while cell division in the pith region occurred mainly between S2 and S3. Therefore, the cortex was established from the PTM at the early development stage of a tuber, with cell enlargement being the main factor responsible for the increases in size of the cortical regions that occurred from S1 to S4. The pith or stele region was established from S2 to S3 by cell division and cell enlargement after the S3 stage.

### 2.2. Transcriptome Analysis of Tiger Nut Tubers

To investigate the molecular mechanism involved in the expansion of the tiger nut, we performed transcriptome analysis of the six stages of tuber development. A total of 18 cDNA libraries were generated. The average clean reads were 24.26 M, the GC content ranged from 48.65% to 49.85%, and the Q20 and Q30 ranged from 93.73–98.31% (Appendix A). Using the tiger nut genome (PRJNA884884) as the reference, the mapped ratio ranged from 86.41 to 88.82%, indicating that the data were reliable and could be used for downstream analysis. Hierarchical clustering analysis was performed to analyze the differentially expressed genes (DEGs) during the development of the tuber and to identify the genes specifically expressed during tuber expansion. The results showed that the 12,472 DEGs could be divided into four clusters. Temporal expression patterns showed that Cluster 1 contained 4696 DEGs, Cluster 2 contained 3219 DEGs, Cluster 3 contained 1870 DEGs, and Cluster 4 contained 2687 DEGs (Appendix A). The expression of DEGs in Cluster 1 were high at the S1 stage and gradually decreased, while DEGs in Cluster 4 showed continuous up-regulation. The DEGs in Cluster 2 showed a trend of first increasing in S2 to S3 and then decreasing (Figure 3A). Based on the relationship between Cluster 1, Cluster 2, and tuber expansion rate, we consider that DEGs in these two clusters mainly regulate tuber expansion.

To further determine the main biological functions of DEGs in the process for tuber expansion, a functional annotation was performed by mapping DEGs to the gene ontology (GO). For biological processes (BP), the main categories were biological regulation, cellular processing, developmental processing, localization, multicellular organic processing, reproductive processing, and responses to stimulation. In the cellular component (CC), the main categories were cell junction, cell periphery, cytoplasm, membrane system, extracellular region, and membrane-enclosed lumen. For molecular function (MF), the main categories were the binding process, catalytic activity, kinase activity, molecular function regulation, motor activity, structural molecule activity, and transcription regulator activity (Figure 3B and Appendix A). Transporter activity and enzyme activity were the most dominant cluster, which indicated that enzyme and transporter activities were important during tuber expansion.

### 2.3. Cell Division-Related DEGs in Tuber Expansion

Plant organ size is usually determined by cell number and size, the rate of cell division, and the number of cells related to the cell division cycle [19]. According to the GO analysis, 780 DEGs were enriched in the cellular process, among which 150 were involved in cell division (GO:0051301) and 252 in the cell cycle (GO:0007049), accounting for 54.5% of the cellular process. These results indicated that cell division and the cell cycle played an important role in tuber expansion (Appendix A).

In order to meet the needs of rapid cell division, genes encoding histone synthesis showed high expression during tuber expansion and included three H1 genes, four H2A genes, 14 H2B genes, 20 H3 genes, and 20 H4 genes. We also showed that seven DNA polymerase genes (POLk, POLε, POLα) had a high expression in the S1–S3 phases than that measured in S4–S6. Rapid cell division is accompanied by rapid nucleotide synthesis and DNA replication [20]. Four adenosine kinase genes (ADK1, ADK2) and two guanylate kinase genes (GK2) related to nucleotide synthesis showed high expression during the tuber expansion phase (Figure 4 and Appendix A).

In the expansion process of a radish root, genes such as cell division protein enzyme (ftsHs), two cell division control protein coding genes (CDC48A), and a cell cycle proteins coding gene (CYCT1) were reported to have a high expression [21]. Cell division, on the other hand, was accompanied by the replication of chromosomes as well as formation of cell plates and changes in membrane lipids. In our study, 19 cyclin (CYC) genes showed a higher expression in S1–S3 phase than that observed in the S4–S6. In addition, 11 cell number regulator (CNR) genes had a 4.47-fold higher expression than that observed in S4–S6 phase. Similarly, five cell division cycle (CDC) genes were more active in the tuber expansion phase. Patellins proteins (PATLs) are a class of lipid transfer active proteins that contain the SEC14 protein domain, which is located on the cell plate of dividing cells and participates in the regulation of plant radicle cell division [22]. In total, 11 PATLs genes showed a high expression in the S1–S3 phase, which may reflect vigorous cell division. Other genes related to cell plate assembly, such as 3 trafficking protein particle complex II-specific subunits (TRS) genes, 3soluble N-ethylmaleimide-sensitive factor adaptor protein 33 (SNAP33) genes, 22 AUGMIN subunits (AUGs) genes were also found to have a high expression during tuber expansion. During the development of tiger nut tubers, cytoskeleton-related genes, including actin (ACT), actin depolymerizing factor (ADF), actin-related protein (ACTR), and actin-related protein 2/3 complex subunit (ARPs) genes had 1.72- to 7.05-fold higher expression than that measured in the S4–S6 phase. Actin binding proteins (ABPs) coding genes, such as four villin (VLN) genes, 17 myosin (MYO) genes, 4 formin homology (FH) genes, 3 fimbrin (FIM) genes, and 3 profilin (PRO) genes had greater than two-fold higher expression in S1–S3 phase compared to that in S4–S6 phase. This suggests that these genes have an important role in the regulation of the depolymerization and the polymerization of microfilaments. As a component of the cytoskeleton [23], tubulin protein (TUB) coding genes showed high expression during tuber expansion and were related loosely to intracellular transport, cell differentiation, cell movement, and cell division (Figure 4 and Appendix A).

### 2.4. Cell Enlargement-Related DEGs in Tuber Expansion

A total of 218 DEGs for cell wall organization or biogenesis (GO:0071554) were enriched during tuber expansion, accounting for 28% of cellular process. These results indicated that the synthesis and metabolism of the cell wall played an important role in tuber expansion. Due to the expansion of the tuber cells, there are many types of transporters on the vacuole membrane. In the development of tiger nut tubers, aquaporin coding genes (6 TIP, 18 PIP, and 5 SIP) had a high transcriptional level in S1–S3 phase. This finding indicated that aquaporin plays a crucial role in maintaining water balance in the expansion phase and is compatible with the high water content and active physiological metabolism of the tubers (Figure 5 and Appendix A).

Cell wall loosening protein coding genes, including 20 expansin (EXP) genes and 11 pectate lyase (PL) genes, showed a high expression in S1–S3 phase. EXP proteins are primary wall loosening agents, with their action being sufficient to restore extensibility to the cell walls. Cellulose microfilaments are the main load-bearing components of the cell wall, and their arrangement determines the direction of cell growth [24]. In total, 3 cellulose synthase like (CSL) genes, 6 xyloglycan-6-xylosyltransferase (XXT) genes, and 20 xyloglucosan endoglycosidase/hydrolase protein (XTH) genes also showed a high expression in S1–S3 phase. A previous study reported that the XTH protein modifies the cellulose xylan composite structure of the cell wall, and it is one of the key enzymes in the process of cell wall reconstruction [25]. Cell wall protein coding genes, including calmodulin (CAM) and calmodulin binding protein (CBP) genes, calcium-dependent protein kinase (CDPK), GPI-anchored proteins (GPI-AP), family member COBRA-like (COBL) and early nodulin-like protein (ENODL93) coding genes showed a high expression in the S1–S3 phase (Figure 5 and Appendix A).

The golgi apparatus is sorted and transported by different transport vesicles in different ways, and there is a large family of proteins involved in vesicle transport. The exocyst complex component plays a key role in the formation of plant cell walls and cell plates. In our study, 33 exocyst complex family (EXO) genes showed a high expression during the tuber expansion phase. Syntaxin family (SYP) genes have been shown to play an important role in mitosis and cell wall construction. Our analyses demonstrated that 27 SYP genes and 5 syntaxin-related protein KNOLLE (KN) genes were more highly expressed during the S1–S3 phase. In total, 20 ADP-ribosylation factors (ARFs) and 16 ADP-ribosylation factor GTPase-activating protein (AGD) genes also had high expression during tuber expansion, while 3 cellulose synthase complex (CSC) genes assembled in the golgi apparatus and localized to the plasma membrane by vesicle transport showed a high expression in the S1–S3 phase (Figure 5 and Appendix A).

### 2.5. Energy-Supply Related DEGs in Tuber Expansion

Sucrose plays a key role in the physiological processes of plant growth, such as energy supply, signal transduction, transcriptional regulation, starch and cellulose synthesis, and stress tolerance. Sucrose reaches the reservoir organ through a series of processes such as phloem loading, long-distance transport, phloem unloading, and distribution among reservoir cells. These processes are dependent on the monosaccharide transporter (MST), sucrose transporter (SUT), and SWEET (sugar will eventually be exported transporter). STP is a sugar transporter in the monosaccharide transporter family, which has a broad spectrum of properties for the absorption of sugar substrates such as glucose, pentose, xylose, ribose, galactose, fructose, and mannose. The transcriptional level of seven STP family genes in the tubers were high in the S1–S3 phase. Sucrose is ultimately decomposed into hexose (glucose and fructose) by cytoplasmic invertase (NINV), vacuole invertase (VIN), or cell wall invertase (CWINV), or alternatively by sucrose synthase (SUS). Sucrose is converted to uridine diphosphate glucose (UDPG) and fructose for the biosynthesis of cellulose and other polysaccharides [26]. In our study, three SUT and eight SWEET genes showed a high expression during the tuber expansion phase. At the same time, 2 sucrose synthase (SUS) genes had 23.82-folder higher expression in the S1–S3 phase than that observed in S4–S6 (Figure 6 and Appendix A). Sucrose is located at the center of carbon assimilation products, which are not only the main end product of photosynthesis carbon assimilation, but also the main transport form for the transportation and distribution of assimilates, thereby playing an important role in energy supply during tuber expansion.

### 2.6. Hormone-Related Pathways Regulate Tuber Expansion

This study screened DEGs involved in biosynthesis and the metabolism pathways for auxin (IAA), cytokinin (CTK), gibberellin (GA), brassinosteroid (BR), and jasmonic acid (JA) during tuber expansion. The three indole-3-pyruvate monooxygenase gene YUCCA5 in the IAA synthesis pathway was highly expressed during the S1–S2 phase. Therefore, the involvement of YUCCA5 in the IAA synthesis pathway may mediate the expansion process of tubers. In the IAA signal regulation pathway, 15 PIN genes and 26 IAA responsive factors (ARF) genes had a high transcriptional level during tuber expansion. IAA1 and IAA21 genes showed a high expression in the S2–S3 phase. It is believed that IAA promotes the tuber expansion process. AtMOB1A promotes IAA signal transduction, with the loss of this function leading to reduced sensitivity to IAA, thereby affecting a variety of developmental processes [27]. Two MOB1A homologous genes of the tiger nut tubers had a high expression in the S1–S3 phase, indicating that MOB1A may mediate tuber expansion by promoting IAA signal transduction (Appendix A and Appendix A).

In the JA synthesis pathway, genes related to lipoxygenase (LOX), allene oxide cyclase (AOC), and allene oxide synthase (AOS) showed high expression during tuber expansion. In the JA signal transduction pathway, two JA signal receptor protein 1a (COI1a) genes, as well as the JAZ protein coding gene, showed high expression in the S1–S3 phase, In addition, the JA response factor, MYC2, also had a high expression during tuber expansion. Taken together, these results indicate that JA promotes the expansion process of tubers (Appendix A and Appendix A). The valine glutamine (VQ) protein, OsVQ13 positively regulates JA signaling by activating the OsMPK6-OsWRKY45 signaling pathway in rice [28]. In this study, two VQ13 homologous genes of tiger nut tubers had high expression in the S1–S2 phase, indicating that they may be involved in the expansion process of pith. The JAZ protein interacts with ABI5, thereby, inhibiting its involvement in the regulation of expression of genes related to the late embryogenesis abundant proteins (LEA). The F-box protein SKP1 interacting partner 31 (SKIP31) interacts with JAZ and then positively regulates the seed maturation process. There is also evidence that ABI5 positively regulates the expression of SKIP31, while JAZ proteins inhibit the regulatory effect of ABI5 on SKIP31 [29]. Two SKIP31 and two ABI5 homologous genes in tiger nut tubers showed an upward trend of expression, which indicated that both SKIP31 and ABI5 positively regulate the tuber maturation process.

Several DEGs were related to the IAA and JA pathways at the tuber expansion stage and, therefore, we next tested their content in tubers at different developmental stages. The results showed that the highest concentration of IAA was found at S1, with IAA content decreasing gradually during tuber development (Figure 7). These endogenous changes in JA indicated that the highest concentration occurred at S1–S3, and decreased from S4. The changes in IAA and JA contents were consistent with the expression patterns of their biosynthetic genes obtained by transcriptome analysis.

In the BR synthesis pathway, a brassinosteroid-6-oxidase (CYP85A1) and two steroid 3-oxidase (CYP90D2) genes were more highly expressed during tuber expansion. Two BR receptor kinases (BRI1) and three BR insensitive 1 (BAK1) genes showed high expression in the S1–S2 phase, while two BR signaling factor (BZR1) genes showed high transcript levels in the S2–S3 phase. OsGSK2 is a GSK3/SHAGGY-like kinase gene homologous to AtBIN2 which negatively regulates the expression of downstream BR responsive genes [30,31]. Six BSK genes showed high expression activity during tuber expansion phase (Appendix A and Appendix A). The IAA-regulated gene involved in organ size (ARGOS) like (ARL) is induced by BR and acts downstream of BRI1 [32,33]. Two ARL genes of tiger nut tubers showed high expression in the S1 stage, a finding which indicated that ARLs regulate the division process in cortical cells. The two IKU1 homologous genes of tiger nut tubers mainly showed high expression in the S1–S3 phase, which may reflect the fact that IKU1 is induced by BR [34,35]. CTK dehydrogenase (CKX) family genes showed a high expression level in the S1–S3 phase (Appendix A and Appendix A). Arabidopsis response regulators (ARRS) are divided into A-type and B-type transcription factors, with the B-type ARR playing a major role in CTK signaling, while the A-type acts as a negative regulator in CTK signaling [36]. During the process of tuber expansion, the A-ARR genes showed a high expression level. Among the DEGs related to GA biosynthesis, two ent-kaurenoic acid monooxygenase (KAO) genes were expressed mainly during the S2–S3 phase. In addition, a GA 3-beta-dioxygenase (GA3ox) gene showed a high expression in the S2–S3 phase. Four GA2ox genes related to the degradation of GA had a high expression in the S1–S3 phase. Therefore, during the process of tuber expansion, the synthesis and degradation of GA reaches a balance (Appendix A and Appendix A).

### 2.7. Transcriptional Pathways Regulate Tuber Expansion

DA1 is a ubiquitin-activated protease, which negatively regulates cell division by shearing the positive regulator of cell division duration [37]. BIG BROTHER (BB) and DA2 activate DA1, DAR1, and DAR2 through monoubiquitination, prompting them to degrade the growth regulators UBP15 and TCPs. There were nine DA1 homologous genes in tiger nut tubers. DA1 and DAR1 were expressed mainly in the S1–S3 phase. At the same time, the transcriptional level of the positive regulators of growth and development during the tuber expansion phase, including two UBP15 and nine BB genes, were significantly higher than that observed in the S4–S6 phase (Appendix A and Appendix A).

In rice, RGG1, GS3, DEP1, and GGC2 positively regulate cell division [38], while RGG2 negatively regulates cell enlargement [39]. Two Gα, one Gβ, and nine Gγ coding genes were found in the tiger nut tubers. The expression of the Gα and Gβ coding genes showed a downward trend during tubers development, while Gγ1 and Gγ3 were expressed at a low level during tuber development, with Gγ2 was expressed mainly at the S1 stage. Three AP2 family transcription factors genes, SMOS1/GR5, showed high expression in the S1–S3 phase (Appendix A and Appendix A).

The phosphorylation of MPK3/6 inactivates DA1, increases the abundance of UBP15, promotes ectodermal cell proliferation, and increases seed size [6]. MAPK6 showed a high expression in the S1–S3 phase. OsMKK10 interacts with and phosphorylates OsMKK4. Activated OsMAPK6 increases grain size by affecting the phosphorylation of downstream transcription factors and promoting the proliferation and expansion of glume cells [40]. MAPKKK10 showed a high expression activity in the S1–S3 phase. OsMKK70 may participate in regulating glume cell proliferation through phosphorylation of OsMKK4 and OsMAPK6, thereby promoting organ enlargement [41]. In addition, OsMKK3 increases grain size by promoting glume cell proliferation [42]. In our study, MAPKKK70 showed high expression activity in the S2–S3 phase during tuber expansion. WRKY53 is activated by the phosphorylation of OsMPK6 and positively regulates glume cell enlargement [43]. Three WRKY53 genes showed a high expression level in the S1–S3 phase in our study, and accordingly may be involved in the regulation of cell enlargement during tuber expansion. OsRAC1 increases the number of cells by promoting the phosphorylation of OsMAPK6, causing an increase in seed width and grain weight [5]. The increased expression of a Rac1 gene was observed in the S1–S3 phase of tuber development (Appendix A and Appendix A).

### 2.8. Transcription Factors in Tuber Expansion

The development process of tubers, including the formation of the cambium meristem and hormone biosynthesis and transportation relies mainly on transcription factors. Our RNA-seq data showed that there were 469 differentially expressed transcription factors during the expansion process of tiger nut tubers, among which 254 belonged to Cluster 1. The characteristic of this type of transcription factor was that they showed a high expression level during the expansion phase (S1–S3) and then gradually decreased with the expression level lower at the later stage of tuber expansion. A total of 215 transcription factors belonging to Cluster 2 were characterized by high expression levels during the rapid expansion phase of tubers (S2–S3). Among these transcription factors, there were 72 bHLH type transcripts, 68 MYB type transcripts, 57 NAC type transcripts, and 48 WRKY type transcripts, they may play an important role in tuber expansion (Appendix A and Appendix A).

### 2.9. Validation of DEGs Using RT-qPCR

To validate the accuracy of the RNA-seq data, qRT-PCR analysis using tubulin beta-4 coding gene TUB4 [44] as a reference showed that 12 genes were involved in cell wall organization and plant hormone signaling (Figure 8 and Appendix A). The results confirmed that the expression profiles of these genes using qPCR were strongly consistent with those obtained from RNA-seq. This finding suggested that the data from the RNA-seq were reliable.

## 3. Discussion

### 3.1. Structure of Tiger Nut Tubers

The storage tissue in most root crops is the secondary xylem and/or phloem, root crops can be divided into two types depending on which tissue, xylem, or phloem demonstrates the growth and storage of nutrients [45]. For example, carrots (*Daucus carota* subsp. *sativus*) [46] form a root crop of a phloem type, while sweet potato [47], cassava (*Manihot esculenta*) [48], and turnip (*Brassica rapa* subsp. *rapa*) [49], form a root crop of a xylem type. There are many differences in the composition of potato, sweet potato, and cassava tubers. Potato tubers are composed mainly of a cortex, perimedullary region and pith, with the perimedullary region accounting for the largest proportion. Sweet potato and cassava tuberous roots are characterized by a central vascular cylinder containing star-shaped xylem, with alternating phloem and parenchyma cells. Like potato tubers, the fully grown tuberous roots of sweet potato and cassava have a comparatively narrow cortex region. The cortex and pith occupy most of the region of tiger nut tubers, with the stele composed mainly of parenchymal cells in the pith, and vascular bundles scattered in the procambium. In contrast to sweet potato tubers, cassava and potato tuberous roots, the cortex in the developing tiger nut tubers occupies a large proportion of the area (Figure 2) and acts as an important storage tissue, similar to that seen in the stele.

### 3.2. Characteristics of Tiger Nut Tuber Expansion

The expansion process of a tuber or root usually results from the division of cambium cells and parenchyma cells in the stele. Root crops can be divided into monocambial or polycambial root crops. The expansion of the single cambium root crops is caused by the division of a cambium ring cell, such as in sweet potato and cassava. The primary vascular cambium of the adventitious roots continues to divide inward to produce secondary xylem and following the division and expansion of cells in the stele, the tuberous roots of sweet potato [50] and cassava [51] expand. The cambium then divides outward to form a small number of cortical cells, with the fully grown tuberous roots having a comparatively narrow cortex region. During the expansion of the radish fleshy root [52], parenchyma cells near the inner side of the cambium parenchyma around the vessels continue to divide, while the number of vessels in the secondary xylem increases. Finally, the xylem becomes completely centered and round and occupies most of the fleshy root area. The thickening growth of polycambial root crops, such as beets (Beta vulgaris), lay down additional cambium rings from pericycle cells, giving rise to the tertiary phloem and xylem [45].

Before potato tuberization, stolon elongation occurs for approximately four days as a result of division of meristem-like cell files along the transversal axis in the stolon tip. The cortex, pith, and regular vasculature can be observed at this stage. Cell division in the stolon tip, then the stop and swelling of the subapical region is initiated with pith cells dividing along the longitudinal axis of the pith and cortex cells until reaching a diameter of 8 mm, at which size tuberization ceases. At this stage, there is limited extension of the outer cortex simultaneity. An extended perimedullary zone with scattered vasculature is then established between the pith and cortex. Further tuber expansion is driven primarily by the expansion of the perimedullary zone until the tuber reaches its final size. However, some tangential cell divisions may still occur in cortical cells that enables further expansion of the tuber [53].

The signs that tiger nut tubers are developing include the expansion of the rhizome tip and the formation of scale-like leaves. Before the tuber expands, the meristem cell region, called the primary thickened meristem (PTM), is surrounded by the pith and cortex at the base of the scale-like leaves primordia and tuber primordia [54]. The PTM are regular, flat, dividing cells arranged near the periphery of the tuber primorda under the leaf primordia, with its continued division responsible for the initial thickening of the developing cortex (i.e., S1–S2 phase) (Figure 2). An increase in the size of the cortex and pith as a result of cell divisions and expansion (S2–S3 phase), results in the PTM differentiating inward into stele cells and outward into cortical cells, which have the characteristics of meristem cells and continue to divide. This is the main reason for the later expansion of tubers (Figure 2). The expansion of tiger nut tubers and potato tubers is, therefore, due to the rapid division of cells. However, the expansion of tiger nut tubers is also due to an increase in cortex thickness caused by the division of PTM cells, whereas the expansion of potato tubers results from the division of pith cells and an increase in the pith size. The subsequent expansion process of potato tubers is attributable mainly to the division and expansion of cells in the perimedullary zone, with the size of the cortex and pith not increasing with the expansion of the tubers. The expansion of tiger nut tubers is driven by the division and expansion of the cortex and pith parenchyma cells. Therefore, in fully grown tiger nut tubers, the cortex and pith occupy the majority of the tuber (Figure 2), while the perimedullary zone occupies the majority of the potato tuber, whereas the cortex and pith occupy only a small area.

### 3.3. Cell Cycle and Cell Enlargement Controls Tuber Expansion

The cell cycle determines the number of cells and regulates cell division. The plant cell cycle is regulated by CYC and CDK [55]. In the development process of the tiger nut tubers, *CYCD1-1*, *CDKB-2,* and *CKS2* genes show a high transcriptional level at the S1 stage, while *CDKG-2* are highly expressed during the S1–S2 phase and are lowly expressed in the S3–S6 phase. Combined with the main events during tuber expansion, these findings indicate that cell cycle-related genes may be involved in regulating the rapid proliferation of cortical cells (Figure 4 and Appendix A).

EXP, as a primary cell wall looseness factor, directly induces enlargement driven by cell wall extension, while XTH, as a secondary cell wall looseness factor modifies the cell wall structure [56]. Combined with the structural changes and gene expression differences during tuber development in tiger nuts, the *EXPA1*, *EXPA10,* and *EXPB8* genes showed a high expression in the S1–S2 phase of development and may be involved in expansion of cortical cells in early tuber expansion. Our study showed that the expression of *EXPA2* was higher in the S1–S3 phase than that observed in the S4–S6 phase and played an important role in the regulation of cell enlargement in both the cortex and stele. The *EXPB6* and *EXPB7* genes showed a high transcriptional level at the S1 stage, which may have been involved in the regulation of cortex cell enlargement (Figure 4 and Appendix A).

### 3.4. The Hormone Signal Transduction Pathway Regulates Tuber Expansion

Plant hormones play a crucial role in regulating the development of abnormal rhizomes [57]. The cell division rate is the highest in the early development of the potato tuber. At this time, the IAA content in the cortex increases by 2.3 times, while the IAA content in the pith increases by 4.7 times [58]. The *YUCCA5* genes were highly expressed in S1–S2, the same as the PIN and AUX genes (Appendix A). This may establish a concentration gradient in the newly formed tiger nut tubers and induce tuber expansion. By analyzing the content of IAA during tiger nut tuber expansion, we showed that the highest concentration of IAA occurred in the S1 stage, and then decreased gradually. According to structural changes, the pith began to differentiate during the S1–S2 phase, indicating that IAA maybe be involved in the differentiation of the stele cells of the tiger nut tubers. In the potato tuber, BRI1 plays a role in regulating tuber size [59]. The expression of the BR synthesis gene *CYP90D2* and signal transduction factor *BZR11* were higher in the S1–S3 phase of tuber development, with the transcription level of the BR receptor *BRI1* gene being increased in the S1–S2 phase (Appendix A). Combined with the changes in structure during the tiger nut tubers development, BR was considered to be involved in the regulation of the whole process of cell proliferation and enlargement in the cortex and pith. There is evidence that the over-expression of *LOX-1*, a key gene for JA synthesis, increases the size of potato tubers [60]. In tiger nut tubers, the JA synthesis related genes, *JMT, LOX5,* and *AOC* showed high expression levels during the S1–S3 phase. *COI1a* and *COI2* were expressed mainly in the S4–S6 phase (Appendix A). At the same time, the changes in endogenous JAs were highest at the S2 stage, and decreased from the S4 stage (Appendix A). These findings suggest that JA is involved in the differentiation of pith cells (Figure 9).

### 3.5. Transcriptional Regulatory Pathways Involved in Tuber Expansion

During the development of tiger nut tubers, we showed that the ubiquitin-proteasome, G-protein, and MAPK signaling pathways were involved in the regulation of the expansion process. OsDA1 directly interacted with OsUBP15 and positive regulator of grain width and size in rice [4]. In tiger nut tubers, DA1 mainly exhibited high transcription at the S2 stage, while BB was expressed mainly in the S3 stage. DA1 plays an important role in promoting the enlargement of the roots of Panax notoginseng [61]. In tiger nut tubers, we consider that DA1 has a role in regulating the proliferation of cortical cells and the differentiation of stele cells, while the BB gene plays a role in the proliferation and enlargement of pith (Appendix A). The Gα and Gβ sub-units positively regulates cell division in rice [38]. This study showed that Gα, Gβ, and Gγ coding genes were highly expressed at the S2 stage and may have promoted the division and enlargement process of cortical cells in tiger nut tubers, as well as initiating the proliferation and differentiation of stele cells. Activated OsMAPK6 increases grain size by promoting glume cell proliferation and swelling [40]. OsMKKK70 also participates in regulating glume cell proliferation by phosphorylating OsMKK4 and OsMAPK6 [42]. WRKY53 can be activated by OsMAKP6 [43]. In tiger nut tubers, MAPKK10, MAPK6, and WRKY53 were highly expressed during the S1–S2 phase, and may have been responsible for regulating the proliferation of cortical cells in the tubers. MAPKKK70 expressed in the S2–S3 phase may also have an important role in the regulation of the differentiation and enlargement of stele cells in tubers (Figure 9). OsRac1 promotes the phosphorylation of OsMAPK6, leading to an increase in seed width and weight [5]. The Rac1 gene exhibited high expression in the S1–S2 phase of tiger nut tubers (Appendix A).

Our study showed that 38 bHLH, 35 MYB, 36 NAC, and 26 WRKY transcription factors had high expression levels at S1 stage, and then showed a downward trend, possibly influencing cortical cell proliferation and stele cell differentiation at the initial phase of tiger nut tuber expansion. In addition, 34 bHLH, 33 MYB, 21 NAC, and 22 WRKY transcription factors showed high expression in the S2–S3 phase of tuber development, and may have played a role in regulating the proliferation and differentiation of pith cells. The function of these transcription factors need to be investigated in greater detail to verify these effects (Appendix A).

### 3.6. Characteristics and Commonalities of Tuber Development in Tiger Nut

The root expansion of single cambium root crop sweet potato [50] and polycambial root crop beets [45] is due to the division and expansion of cambium cells, while the inward differentiation of PTM into stele cells and the outward differentiation into cortical cells led to the expansion of the tiger nut tubers. The perimedullary region accounted for the largest proportion in potato tubers [53], as well as sweet potato and cassava, the fully grown tuberous roots have a comparatively narrow cortex region [50,51]. Unlike potato, sweet potato, and cassava, the cortex development preceding stele development in tiger nut tubers, and the cortex and stele occupies most of the region of tubers. In addition, several signaling pathways regulating plant organ cell division and expansion were also found to regulate the expansion process of tiger nut tuber [3,4,5,6,7,8,9,10,11,12,13], the auxin signaling pathway promotes the division of cortical cells; the jasmonic acid pathway, brassinosteroid signaling, G-protein pathway, MAPK pathway, and ubiquitin protein pathway regulate cell division and the expansion of the tuber cortex and stele (Figure 9).

## 4. Materials and Methods

### 4.1. Planting and Sampling

The tiger nut strain “B63” was selected for this study. The tubers were soaked in water for two days to promote maximum vitality and uniform germination, and then placed in 30 × 30 cm (diameter × height) of cloth bags containing a soil mixture (nutrient soil: vermiculite, 1:1). The tubers were sown at a depth of 2.0 cm. Under natural sunlight conditions, seed tubers germinate in moist soil and grow to maturity. After sowing, the tubers were collected at six different developmental stages, i.e., day 40 (S1), day 50 (S2), day 60 (S3), day 70 (S4), day 90 (S5), and day 110 (S6). At least three independent biological replicates were collected for each stage. All tissues were immediately stored at −80 °C for subsequent analysis.

### 4.2. Preparation of Paraffin Sections

The tubers were fixed with formaldehyde, alcohol, acetic acid (FAA) fixative, dehydrated in gradient alcohol solutions and then made transparent using anhydrous ethanol and xylene. The samples were soaked in wax for 60 min, with the soaking steps repeated three times, embedded using an embedding machine, and then sliced. The slices were soaked in xylene for 30 min, with the process being repeated two times. The slices were then soaked in an anhydrous ethanol/xylene solution (anhydrous ethanol:xylene = 1:1) for 30 min to dewax. After dewaxing, the slices were dyed with 1% safranine solution for 10 h, decolorized in gradient alcohol solutions, and then dyed in 0.5% solid green dye solution for 30 s. After dehydration with anhydrous ethanol, the slices were dried and sealed with gum. The changes in the tissues and cells were observed and photographed using an ordinary optical microscope [62].

### 4.3. Determination of Hormone Content

The tuber samples were selected, ground in liquid nitrogen, and then placed in a glass tube, followed by the addition of a mixture of isopropanol, water, and hydrochloric acid. An 8 µL aliquot of a 1 µg/mL standard solution was then added, followed by shaking for 30 min, addition of dichloromethane and further shaking for 30 min. After centrifugation at 13,000 rpm for 5 min, the lower organic phase was collected avoiding the introduction of light. The organic phase was then dried with nitrogen and redissolved in methanol (0.1% formic acid). After centrifugation at 4 °C for 10 min (13,000 rpm), the supernatant was collected, and the hormone content detected by HPLC-MS/MS. Briefly, the steps of this analysis were as follows: (1) the preparation of a standard sample solution and establishment of a quantitative working curve, (2) the preparation of a sample solution and measurement of the recovery rate, and (3) the calculation of the IAA and JA contents [63].

### 4.4. RNA Extraction, cDNA Library Construction, and RNA-Seq

Tubers collected at the S1, S2, S3, S4, S5, and S6 stages were chosen for transcriptome sequencing according to the characteristics and dynamic changes in total oil content during development of the tiger nut tubers. Total RNA was extracted using TriZol reagent according the manufacturer’s instructions (Invitrogen, USA) and genomic DNA then removed by DNase I (TaKaRa, Japan) digestion. The transcriptome library was prepared using the TruSeq RNA sample preparation kit (Illumina, San Diego, CA, USA) using 5 μg of total RNA, which was then sequenced with Illumina HiSeq 4000 (2 × 150 bp read length) supplied by Biomarker Technologies Co, Ltd. (Beijing, China). For gene expression analysis, the number of expressed tags was calculated and then normalized to TPM (Trans per kilobase of exon model per million mapped reads) [64].

### 4.5. Transcriptome Data Analysis

The sequencing data were mapped to the genome sequence using Hisat 2 software. The data between different samples were compared, and the raw data then standardized to detect the DEGs. The relative expression level value of the DEGs was calculated by reading the TPM value, and finally the corrected *p*-value. The false discovery rate, (FDR), was adopted as the key index for screening of the DEGs. All the genes were subjected to clustering analysis using the Mfuzz package [65]. Gene ontology (GO) enrichment analysis of the DEGs was performed using the cluster Profler R package [66].

### 4.6. qRT-PCR Verification

Real-time (qRT-PCR) was performed using a Promega kit (Go Taq^®^ qPCR Master Mix) and the Light Cycler 480 system, with the GoScriptTM reverse transcription system (Promega, Madison, WI, USA) reverse transcription cDNA used as the template [67]. The primers were designed using Primer Premier 6 software. The reaction system was 20 μL in volume and consisted of 10 μL 2 × NovoStart^®^SYBR, 10 μL of qPCR SuperMixPlus, 1.0 μL of each primer, 7.0 μL of RNase-free water, and 1 μL of template. The PCR cycle was 40 cycles of 95 °C, 10 min; 95 °C, 5 s; 62 °C, 20 s; 72 °C, 20 s. The ΔΔC_T_ method for relative gene quantification was used to calculate the expression level of each gene in the six stages of tuber development. Three biological replicates were used for the qRT-PCR analysis.

## 5. Conclusions

This study combined transcriptome analysis and anatomical and physiological observations to identify the candidate genes that regulated cortical development and pith in tiger nut tubers and were involved in cell wall development and the cell cycle, plant hormone signal transduction, MAPK signal transduction, the G-protein-mediated signal pathway, and the ubiquitin-mediated signal pathway. Finally, we proposed a hypothetical model for the genetic regulatory network for tuber expansion in the tiger nut. The expansion of tiger nut tubers was shown to be due mainly to cell differentiation, division, and expansion, which were regulated and promoted by specific signal transduction pathways and metabolic processes. These studies not only provide a new concept for the molecular regulation mechanism of tuber expansion, but also provide a theoretical basis for genetic improvement of tiger nuts.

## Figures and Tables

**Figure 1 ijms-25-05100-f001:**
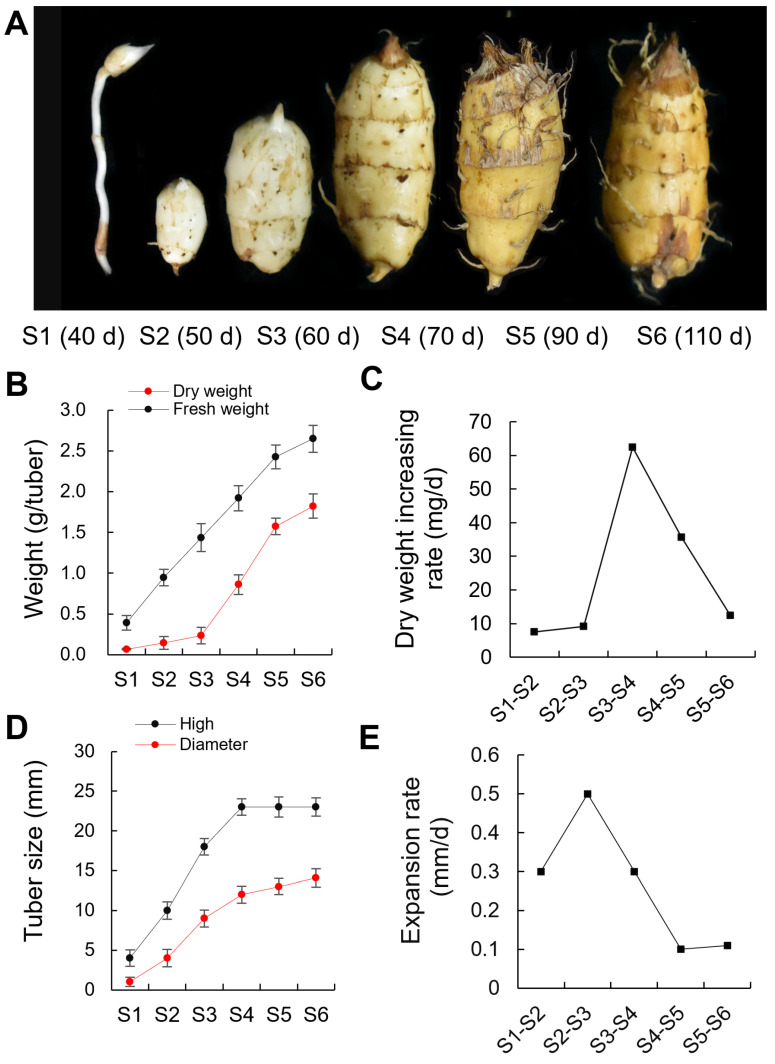
Characterization of tiger nut tubers development. (**A**) Morphological changes in the developing tubers. (**B**) Changes in the dry and fresh weight per tuber during tuber development. (**C**) The rate of the dry weight increases during tuber development. (**D**) Changes in the tuber size during tuber development. (**E**) Daily tuber expansion rate during tuber development. The horizontal axis shows the six stages of tiger nut tubers development. S1, 40 days after sowing (40 d); S2, 50 d; S3, 60 d; S4, 70 d; S5, 90 d; and S6, 110 d.

**Figure 2 ijms-25-05100-f002:**
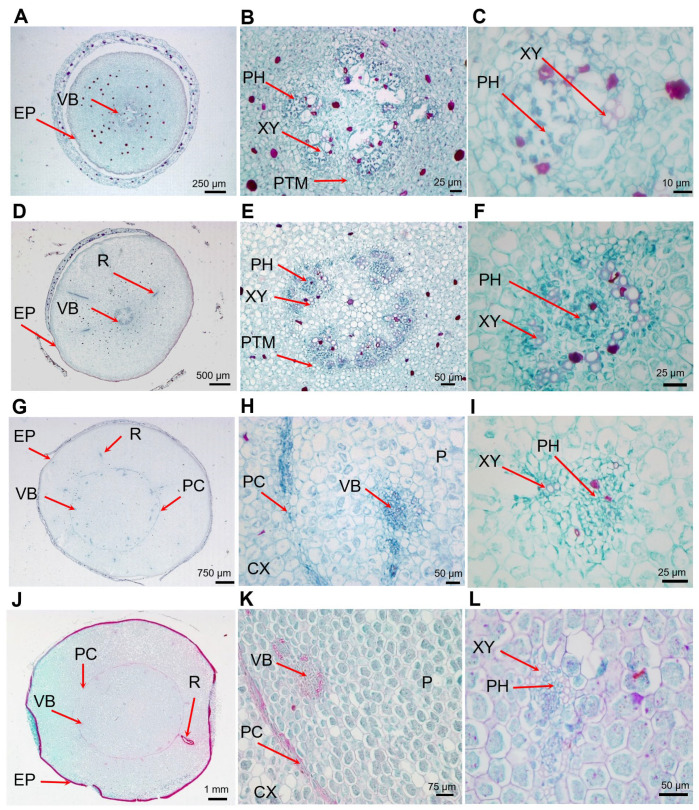
Cross-sectional photographs of the developing tubers. (**A**) Cross-section of a stage 1 (S1) tuber. (**B**) Cross-section of an S1 tuber showing the stele and PTM. (**C**) Cross-section of an S1 tuber showing the vascular bundle. (**D**) Cross-section of an S2 tuber. Adventitious roots have generated and the cortex occupies most of the area of tuber. (**E**) Cross-section of an S2 tuber showing the stele and PTM. (**F**) Cross-section of an S2 tuber showing the vascular bundle. (**G**) Cross-section of an S3 tuber. (**H**) Cross-section of an S3 tuber showing the PC and VB. (**I**) Cross-section of an S3 tuber showing the PH and XY in the VB. (**J**) Cross-section of an S4 tuber. (**K**) Cross-section of an S4 tuber showing the PC and VB. (**L**) Cross-section of an S4 tuber showing the PH and XY in VB. CX, cortex; EP, epidermis; P, pith; PC, procambium; PH, phloem; PTM, primary thickening meristem; R, adventitious roots; VB, vascular bundles; XY, xylem.

**Figure 3 ijms-25-05100-f003:**
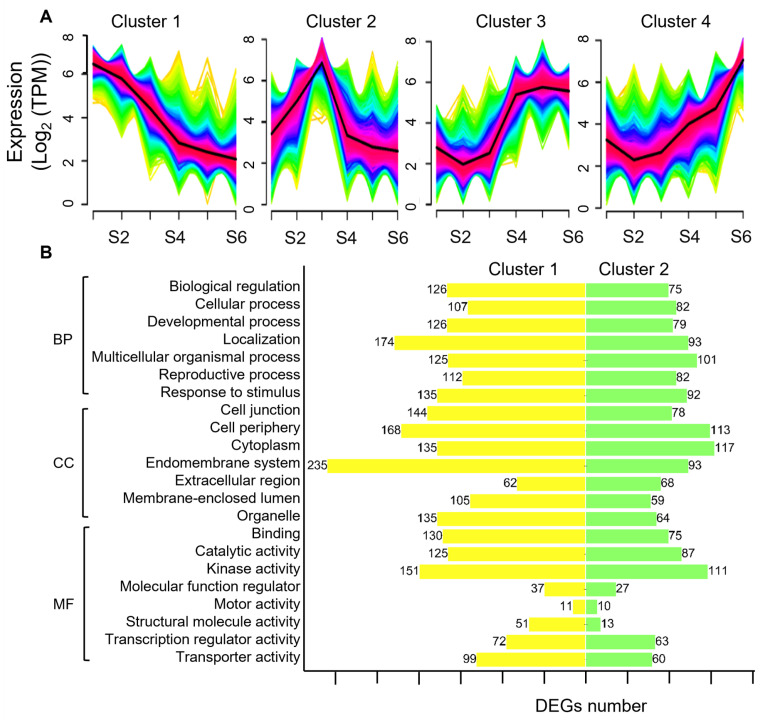
Clustering and the functional analysis of differentially expressed genes. (**A**) Clusters of the differentially expressed genes. The horizontal axis shows the six stages of tiger nut tubers development; S1: 40 days after sowing (40 d), S2: 50 d, S3: 60 d, S4: 70 d, S5: 90 d, and S6: 110 d. (**B**) Classification of the differentially expressed genes in Clusters 1 and 2, based on GO clustering. BP, biological process; CC, cell component; MF, molecular function.

**Figure 4 ijms-25-05100-f004:**
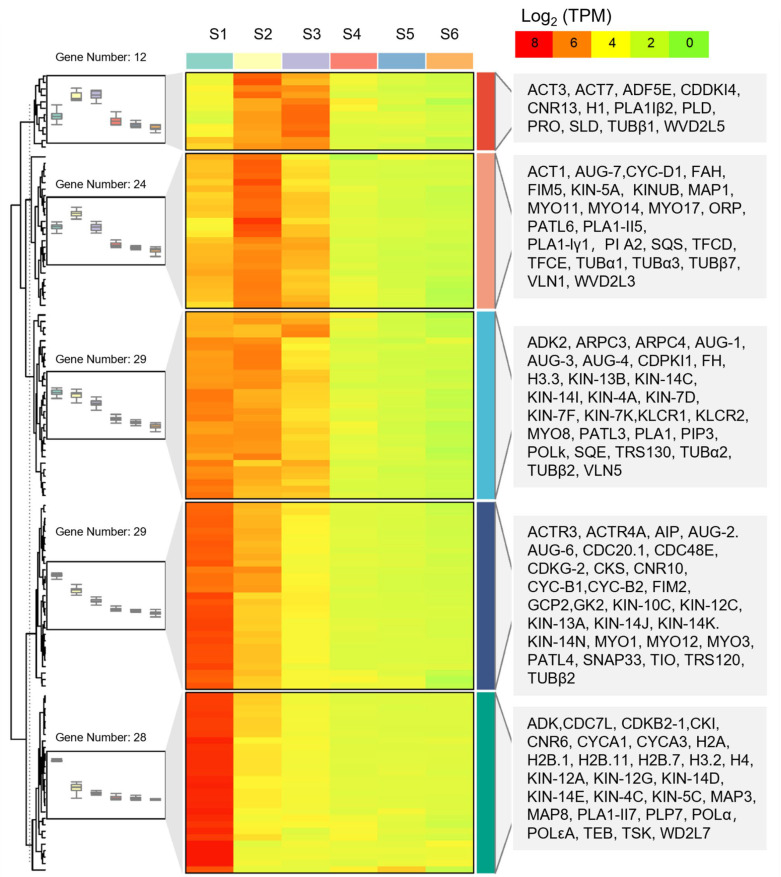
Hierarchical cluster analysis of differentially expressed genes related to cell division during tuber expansion. The box-plot shows the patterns of genes expression, the heat map shows gene expression, the right side lists the names of the genes. The heat map was drawn based on log_2_ (TPM) values. The red bands indicate a high gene expression and the green bands indicate a low gene expression.

**Figure 5 ijms-25-05100-f005:**
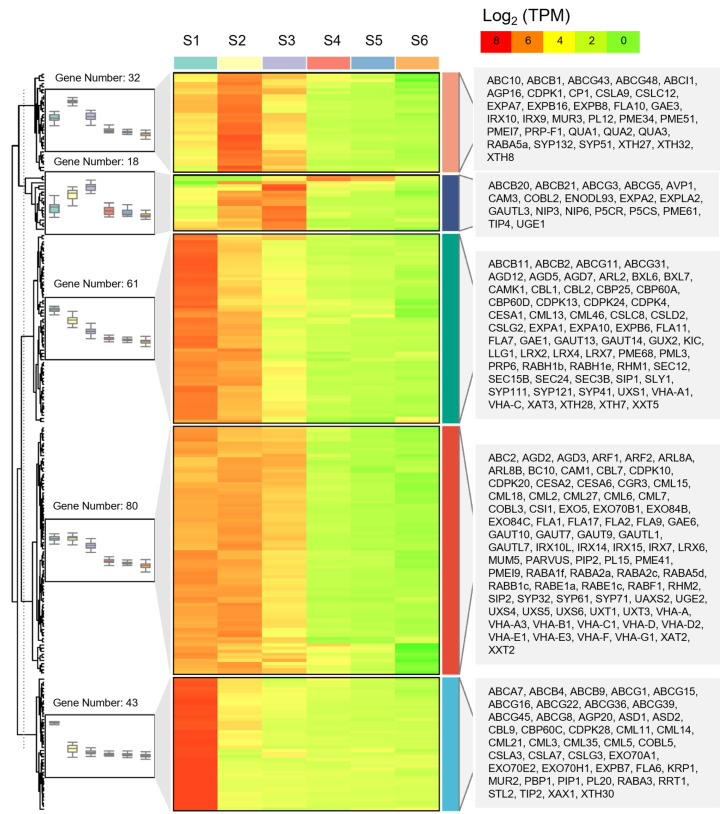
Hierarchical cluster analysis of differentially expressed gene related to cell enlargement during tuber expansion. The box-plot shows the patterns of gene expression, the heat map shows gene expression, the right-hand list shows the names of genes. The heatmap was drawn based on log_2_ (TPM) values. The red bands indicate a high gene expression and the green bands indicate a low gene expression.

**Figure 6 ijms-25-05100-f006:**
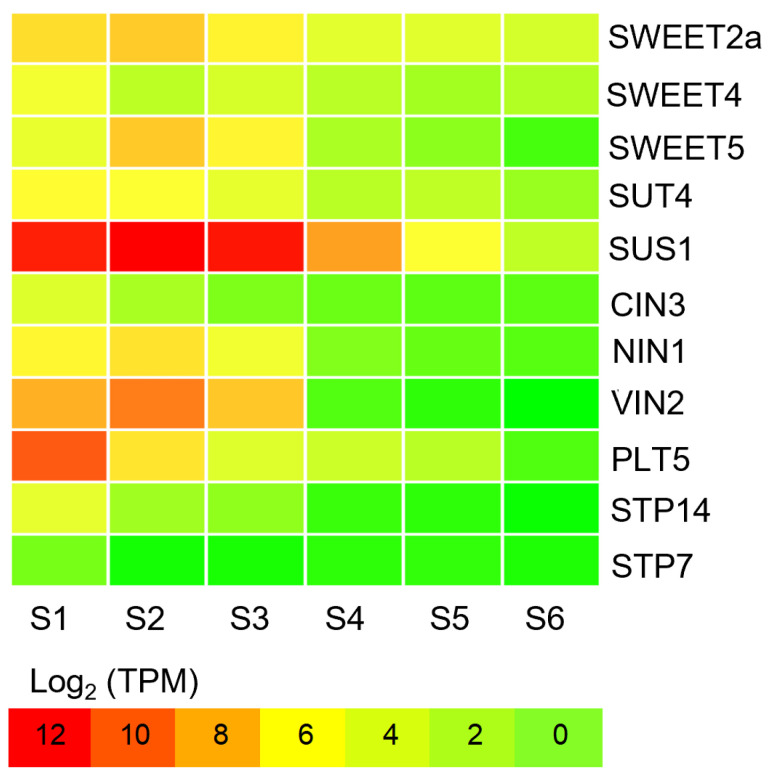
Heatmaps showing differentially expressed genes related to energy supply during tuber development. The horizontal axis shows the six stages of tiger nut tubers development; S1, 40 days after sowing (40 d); S2, 50 d; S3, 60 d; S4, 70 d; S5, 90 d; and S6, 110 d. Information on the genes is detailed in Appendix A. The heatmap was drawn based on log_2_ (TPM) values. The red bands indicate a high gene expression and the green bands indicate a low gene expression.

**Figure 7 ijms-25-05100-f007:**
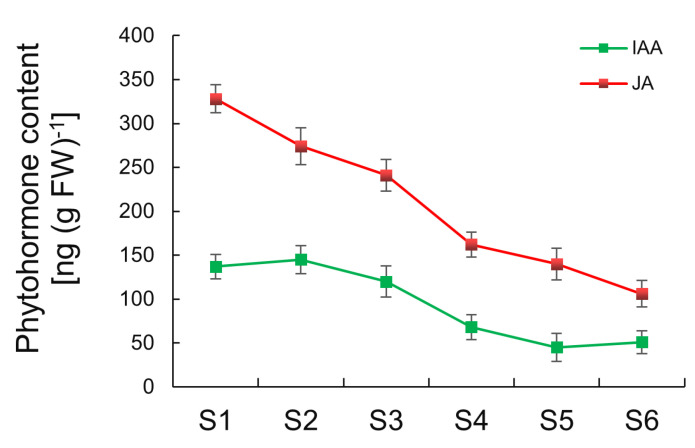
The content of endogenous hormone during tuber development. S1, 40 day after sowing (40 d); S2, 50 d; S3, 60 d; S4, 70 d; S5, 90 d; and S6, 110 d. The data are expressed as the mean ± SD (*n* = 3). The red line shows the change in auxin (IAA) and the green line indicates the change in jasmonic acid (JA) in the tubers.

**Figure 8 ijms-25-05100-f008:**
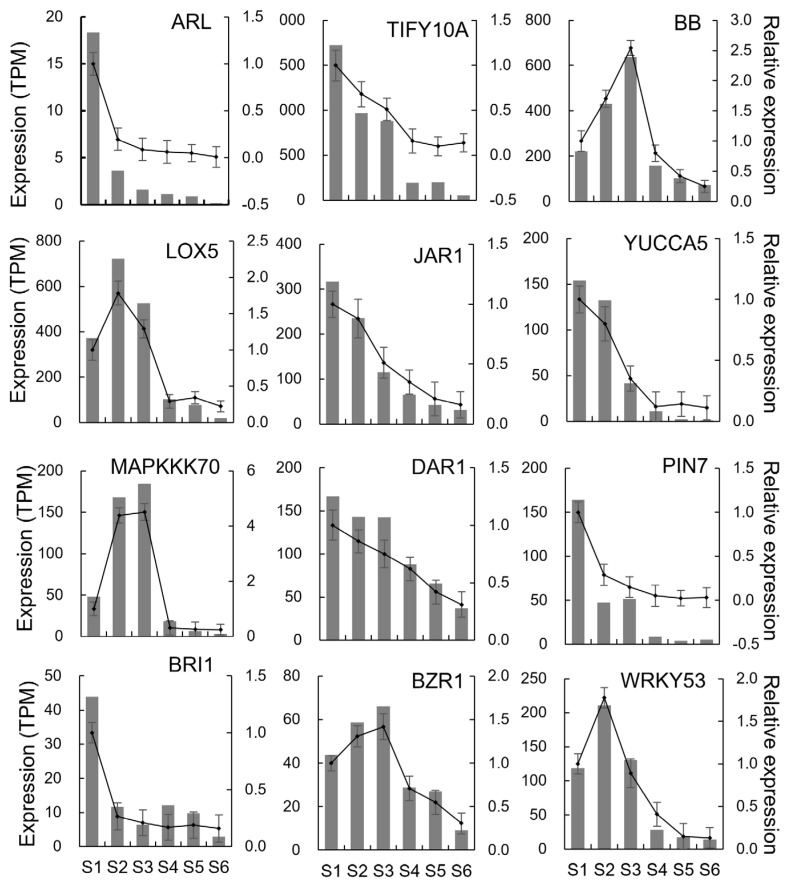
Relative changes in gene expression measured by RT-qPCR. S1, 40 day after sowing (40 d); S2, 50 d; S3, 60 d; S4, 70 d; S5, 90 d; and S6, 110 d. The left ordinate represents the gene expression data from RNA-seq and the right ordinate represents the relative gene expression data obtained from RT-qPCR using the gene expression of S1 stage as a control. The black column is RNA-seq data and the black line is relative expression data. The values are expressed as the mean ± SD (*n* = 3). The detailed sequence information of the primers used is shown in Appendix A.

**Figure 9 ijms-25-05100-f009:**
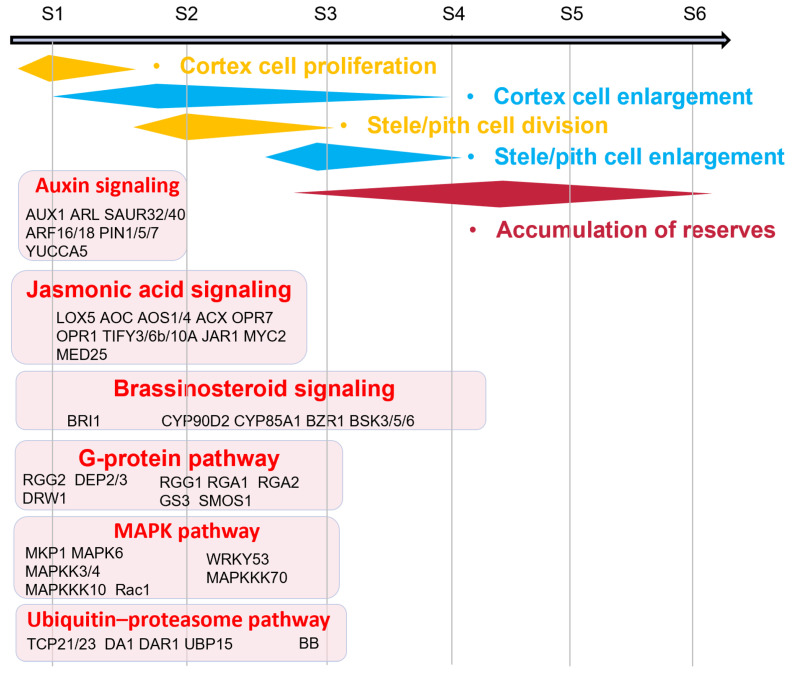
Regulatory pathway of the tuber expansion in tiger nuts. The different processes occurring during tuber development (cell division and cell enlargement) are represented. S1, 40 days after sowing (40 d); S2, 50 d; S3, 60 d; S4, d; S5, 90 d; and S6, 110 d. The auxin signaling pathway mainly functions during S1–S2; the jasmonic acid pathway, brassinosteroid signaling, G-protein pathway, MAPK pathway, and ubiquitin protein pathway regulate cell division and the expansion of the tuber cortex and stele. Detailed information on these genes is shown in Appendix A.

**Table 1 ijms-25-05100-t001:** Cell size of the tubers at different development stage.

Stage	Cell Size (μm)				Cortical Thickness (mm)	Stele Radius (mm)	Cortex Thickness:Stele Radius Ratio
	Cortex		Stele				
	Tangential	Radial	Tangential	Radial			
S1	21.47 ± 2.83	15.26 ± 2.19	15.21 ± 1.69	10.49 ± 1.21	0.70 ± 0.01	0.17 ± 0.02	3.27
S2	42.09 ± 6.10	28.77 ± 4.62	26.89 ± 2.86	19.40 ± 2.46	1.39 ± 0.01	0.23 ± 0.02	5.04
S3	50.31 ± 5.32	36.93 ± 4.83	52.74 ± 11.73	38.57 ± 8.89	1.85 ± 0.01	1.97 ± 0.01	0.90
S4	56.32 ± 9.74	39.59 ± 11.31	60.47 ± 9.08	50.43 ± 11.59	2.39 ± 0.02	2.70 ± 0.02	0.88

Note: 200 cells were counted, with the data expressed as mean ± SD of three independent biological replicates.

## Data Availability

The data presented in this study are available in this article and Appendix A.

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
