# Peer review of "RNA-Seq Reveals That Multiple Pathways Are Involved in Tuber Expansion in Tiger Nuts (Cyperus esculentus L.)"

_ijms, 2024, doi:10.3390/ijms25105100_

Round 1

Reviewer 1 Report

Comments and Suggestions for Authors

The manuscript of Hou G et al. “RNA-seq Reveals Multiple Pathways are Involved in Tuber Expansion in Tiger Nut (Cyperus esculentus L.)” dedicated to clarifying the anatomical and molecular mechanisms of tuber expansion in tiger nut plants. The authors present the results of a comprehensive study, including morphometric and microscopic analysis, determination of hormone content, determination of transcriptomes and their bionformatic analysis as well as qRT-PCR verification. Eventually, the authors proposed an interesting hypothetical model for the genetic regulatory network for tuber expansion in the tiger nut. As a whole, the paper is well done and well written.

Author Response

Dear Editor,
Thank you very much for your affirmation of our work, and we look forward to subsequent cooperation.

Reviewer 2 Report

Comments and Suggestions for Authors

Dear Authors,
Your manuscript titled „RNA-seq Reveals Multiple Pathways are Involved in Tuber Expansion in Tiger Nut (Cyperus esculentus L.)” contains very interesting results. Nevertheless, I have found some imperfections, which (in my opinion) should be corrected or at least clarified before an eventual publication in IJMS. I have listed them below:
1.    I suggest to add main consclusion in section Abstract.
2.    At the end of chapter Introduction the main or/and specific goals of investigations should be presented.
3.    Lines 99-101 in my opinion the more detailed characteristics of Cyperus esculentus should be presented. Such characteristics should contain information about range of taxon, habitat affiliation, lifespan of individuals, mode of reproduction (generative and vegetative) allowing the spreading of individuals).
4.    I suggest to add at least brief chapter Discussion, which might contain the comparisons of obtained extent outcomes with literature of subject refering to other tuberous taxa. This section should highlight the novelty of results presented by Authors of manuscript.

Author Response

Comments 1: I suggest to add main consclusion in section Abstract.

Response 1: We have added the following sentence in lines 31-35 of the manuscript: Finally, we come to this conclusion that the cortex development preceding stele development in tiger nut tubers. The auxin signaling pathway promotes the division of cortical cells, while the jasmonic acid pathway, brassinosteroid signaling, G-protein pathway, MAPK pathway, and ubiquitin protein pathway regulate cell division and expansion of the tuber cortex and stele.

Comments 2: At the end of chapter Introduction the main or/and specific goals of investigations should be presented.

Response 2: We have added the following sentence in lines 105-108 of the manuscript:

Clarifying the mechanisms underlying tubers development is important for improving the tiger nut yield. Therefore, based on the observation of tuber anatomy and transcriptome sequencing during tuber development, the regulatory mechanism of tuber enlargement was discussed in this paper. 

Comments 3: Lines 99-101 in my opinion the more detailed characteristics of Cyperus esculentus should be presented. Such characteristics should contain information about range of taxon, habitat affiliation, lifespan of individuals, mode of reproduction (generative and vegetative) allowing the spreading of individuals).

Response 3: We have added the following sentence in lines 90-92 of the manuscript:

Tiger nut is a grass-like perennial herb that belongs to the sedge family Cyperaceae that is widely distributed in tropical, subtropical and temperate zones, it can be used as an annual cultivated crop in agricultural production [14].

Comments 4: I suggest to add at least brief chapter Discussion, which might contain the comparisons of obtained extent outcomes with literature of subject refering to other tuberous taxa. This section should highlight the novelty of results presented by Authors of manuscript.

Response 4: We have added the following sentence in lines 609-623 of the manuscript:

3.6 Characteristics and Commonalities of Tuber Development in Tiger nut

The root expansion of single cambium root crop sweet potato [50] and polycambial root crop beets [45] is due to the division and expansion of cambium cells, while the inward differentiation of PTM into stele cells and the outward differentiation into cortical cells led to the expansion of the tiger nut tubers. The perimedullary region accounting for the largest proportion in potato tubers [53], as well as sweet potato and cassava, the fully grown tuberous roots have a comparatively narrow cortex region [50, 51]. Unlike potato, sweet potato and cassava, the cortex development preceding stele development in tiger nut tubers, and the cortex and stele occupy most of the region of tubers. In addition, several signaling pathways regulating plant organ cell division and expansion were also found to regulate the expansion process of tiger nut tuber [3-13], the auxin signaling pathway promotes the division of cortical cells; the jasmonic acid pathway, brassinosteroid signaling, G-protein pathway, MAPK pathway, and ubiquitin protein pathway regulate cell division and expansion of the tuber cortex and stele  (Figure 9).